# OpenReview forum: "The Curse of Low Task Diversity: On the Failure of Transfer Learning to Outperform MAML and their Empirical Equivalence"
_ICLR.cc/2023/Conference — Submitted to ICLR 2023_

### Official Review · Reviewer_NZ7N · 2022-10-24

**Confidence:** 4
**Clarity, Quality, Novelty And Reproducibility:** See above
**Correctness:** 1
**Technical Novelty And Significance:** 3
**Empirical Novelty And Significance:** 2
**Recommendation:** 3

**Strength And Weaknesses:**

Overall, I appreciate the thoroughness in the execution and presentation of the experiments in this work. The ablation over multiple probe networks lend credence to the robustness of the proposed metric over the choice of the probe network. The hyperparameters and training details of each model used in the experiments are well documented. As such, the paper states convincingly that the performance of MAML and transfer learning are statistically equivalent on MiniImagenet and Cifar-fs.

The writing quality of the paper is fairly strong. The authors went into great detail to explain background works, including formulations for the feature similarity metrics, in the appendix, which makes the work nicely self-contained. There is no noticeable ambiguity in any of the formulations/algorithms used in the paper. The proofs regarding similarity of a MAML-type algorithm and a “fixed representation meta-learner without adaptation” are correct, albeit straight-forward.

There are some problems that I notice in this work, which I list in the order of severity.

1. The claim that MiniImagenet and Cifar-fs has low diversity is stated without context. While indeed the proposed diversity coefficient can range from 0 to 1, it is not clear where do real meta-learning datasets actually lie on this 0-to-1 scale. Without this context, one cannot claim that a value of 0.1 or 0.06 should be considered “low”.

2. The authors suggest that their experiments point towards the following conclusion: “if the task diversity is low, then transfer learned solutions might fail to outperform meta-learned solutions”. In close examination, this claim is in fact vacuous (or unsupported, depending on the interpretation of if). Even if we assume the claim regarding “low” diversity to be correct, the experimental results (on real datasets) in this paper only indicates that transfer learned solutions fail to outperform meta-learned solutions in “low” diversity tasks, but this could be the case regardless of whether task diversity is low. The observations have only shown co-occurrence (“roses are red, sky is blue”), but a non-vacuous claim at least requires correlation (“roses are more red when sky is blue”), which requires negative examples (“roses are not red, sky is not blue”).

3. The synthetic experiment in this paper may refute the above issue as it shows transfer learning out-performing MAML (the Gaussian experiment) in “high” task diversity settings. Yet, the control variable (task diversity) has a significant confounding variable that could directly explain the observation (USL beating MAML), the variable being task difficulty. The authors adjust the variance of the distribution of class means (sigma_m) to create a range of benchmarks with different diversity values. However, increasing the variance of class means also increases the average distance between the class means, which in turn increases the margin in classification boundary between each class for any given 5-way classification episode, thereby making the classification problem easier. This can be easily observed in figure 3, where the meta-test accuracy of both methods increases consistently with increased diversity, indicating a decrease in difficulty. We can then postulate that USL out-performs MAML in the 3 benchmarks on the right-end of the graph not due to diversity, but due to difficulty. A likely explanation is that the L-BFGS algorithm used for optimizing the last layer is fully converged, while MAML-5’s 5 gradient steps are not fully converged. When the classes are so well separated (e.g. when inter-class variance is 20 times that of intra-class variance), a linear model (i.e. last layer of a mlp) can be trained to convergence on very few examples without fear of overfitting, as the likelihood of outliers is vanishingly small. This explanation predicts that using L-BFGS on the last layer with the feature-extractor learnt by MAML should close the performance gap, as the difference is in the meta-test time optimization, not MAML vs USL. In addition, this experiment highlights the fact that the proposed diversity metric is sensitive to task difficulty, which should be an unwanted side effect. The authors should conduct experiments in settings where the diversity varies while the difficulty is controlled, to illustrate how MAML and USL responds to various diversity levels.
4. While the authors do not make any strong claims w.r.t their theoretical analysis, I want to point out that the analysis misses the point by analyzing the learner trained by equation 12 instead of equation 13. Without any task adaptation or conditioning, it is rather obvious that the decision rule can only learn the mean prediction over all tasks (as pointed out by the authors themselves). This decision rule is entirely useless, as it will only learn trivial solutions when the assignment between classes and labels is shuffled (e.g. all standard benchmarks). On the other hand, the result of when equation 11 is equivalent to 12 in the zero task diversity setting is weak in its current form - a much more useful result would be to show that the two learners tend towards the same when diversity converges to zero.



**Summary Of The Paper:**

This work proposes a novel diversity metric that measures the dissimilarity between few-shot learning tasks. The authors find that popular few-shot learning benchmarks - MiniImagenet and Cifar-fs, score low on the proposed diversity metric. The authors then compare the performance of MAML to transfer learning in these two benchmarks and find that under standardized hyperparameters, MAML and transfer learning are statistically similar in performance across many model configurations. Thus, the authors conclude that the similarity in performance between MAML and transfer learning is largely due to the low task diversity.


**Summary Of The Review:**

In balance, I think the paper does not make a sufficiently novel claim to meaningfully contribute to our collective knowledge, thus I recommend rejection.

---

### Official Review · Reviewer_nDz4 · 2022-10-24

**Confidence:** 4
**Correctness:** 2
**Technical Novelty And Significance:** 2
**Empirical Novelty And Significance:** 2
**Recommendation:** 1

**Clarity, Quality, Novelty And Reproducibility:**

The paper is well organized. However, certain critical aspects are missing, or often relegated to the supplementary material. For example, MAML5 and MAML10 are defined in the caption for Figure 1, but appear in Table 1.

There are a few typos - Figure 2 caption MAM15 - MAML5.


**Strength And Weaknesses:**

Strengths

The earlier work by Raghu et al, did compare MAML against multi-class (USL) and multi-task learning. However, their analysis was restricted to studying the quality of features learned through different training regimes. The key difference between Raghu etal’s work and this paper is the training of an additional classifier (adapt head only). The feature extraction backbone is not very influential in the low-task diversity setting as logistic regressors trained using MAML and USL feature extraction backbones result in comparable performance. This is an interesting observation and main strength of the paper (though I have some reservations as elucidated later)

[Raghu et al] Aniruddh Raghu, Maithra Raghu, and Samy Bengio. Rapid Learning or Feature Reuse? Towards Understanding the Effectiveness of MAML. Technical report, 2020. URL https://arxiv. org/abs/1909.09157.

The diversity metric based on the Hellinger distance of the Task2Vec representations is quite intuitive and sound.

The authors conduct experiments using benchmark datasets for few-shot learning. The variety of distance metrics such as SVCCA, PWCCA, and the likes reinforces the observations from the earlier experiments.

Weaknesses

The diversity metric is computed using only the Resnet backbones (Table 1). I wonder if the similarity of the backbone architecture is a cause for the similar diversity results? Experimenting with diverse architectures (perhaps VGG and densenet variants) might strengthen the observation.

My primary concern is the task that the authors have used for quantifying diversity. The results in Table 1 and 2 are from 5-way 5-shot tasks (though I understand that 20 shots were used to estimate the diversity score in Table 1). I believe that a 5-way task is insufficient to draw out the diversity across tasks, especially when the tasks are sampled from miniimagenet or CIFAR-FS datasets. I would urge the authors to try out 10-way or even higher N-way tasks, with the same number of shots. If the authors were keen to stick to a 5-way task, I would have expected results from a 5-way 1-shot setting as well. The current set of results, though suggestive of a trend, are not convincing enough to draw an inference.

I like the experiment on increasing the model size as a function of the number of filters. The results are presented in Figure 2. However, here too, I find that the authors have stopped the experiment at a critical juncture. I would have liked to see runs beyond 30 filters. There is an increasing trend in favor of USL. This should ideally get amplified with larger model size. It is difficult to conclude by stopping at 30.

I also find results presented in Figure 3 to be quite strange. How can the meta-test accuracy increase with higher Task2Vec diversity coefficient? One would have expected the accuracy to actually reduce with harder tasks, the current trend seems to be quite counter intuitive as both MAML and USL have higher accuracies on difficult tasks. Further, similar to the previous observation, the authors stop at 0.5, when things start to get interesting - the difference between MAML and USL is apparent here. I would urge the authors to continue further (as indicated in the last subplot of Figure 3).

While using synthetic data helps to control task diversity, I would urge the authors to try a more natural setting of higher task diversity - cross domain-few shot learning tasks. This should give better insights into behavior of both MAML and USL.


**Summary Of The Paper:**

Model agnostic meta-learning (MAML) and transfer learning (TL) are popular techniques for learning under data scarcity such as  few-shot learning. While the effectiveness of MAML over TL has been shown for medical image analysis tasks, the verdict on classical benchmark datasets in this domain is still unclear. This paper presents a systematic study comparing MAML and TL under fair conditions (architecture, optimizer, etc.) across diverse tasks. The authors introduce a diversity coefficient to measure task diversity using Task2Vec representations. Using this measure, the authors show that TL and MAML results are comparable outcomes when the task diversity is low. This observation holds even as the model size changes.


**Summary Of The Review:**

Overall, the paper is an interesting attempt to contextualize the behavior of MAML and USL algorithms. While I agree that task diversity is an important metric for evaluation, I find the work needing additional experiments. There are interesting patterns in the results, but I am afraid the experiments are not complete enough to draw conclusions.

---

### Official Review · Reviewer_NP6p · 2022-10-24

**Confidence:** 4
**Correctness:** 3
**Technical Novelty And Significance:** 2
**Empirical Novelty And Significance:** 2
**Recommendation:** 3

**Clarity, Quality, Novelty And Reproducibility:**

- Clarity:
    - The paper reads mostly OK but doesn’t feel quite polished. See my remarks on presentation above.
- Quality:
    - This paper could also be improved in terms of quality, by discussing prior work and comparing to it.
- Novelty:
    - This is the biggest flaw of the paper: while the diversity coefficient is arguably novel (Achille and collaborators have also used Task2Vec to measure the distance across tasks), it is used to provide insights that are already known to the community.
- Reproducibility:
    - The authors use open-source software, and I believe most of their results are reproducible.

**Strength And Weaknesses:**

- Strengths:
    - The use of Task2Vec to compute diversity is novel in the context of few-shot classification (but Achille et al. already used it for similarity across vision tasks).
    - I appreciated the experiments on the toy Gaussian experiments: it clearly shows a strong correlation between the proposed diversity coefficient, ground truth Hellinger distance, and the meta-test accuracy.
- Weaknesses:
    - More thorough validation: while the paper focuses on MiniImageNet and CIFAR-FS (which are well-known to have similar train and test tasks), I wonder if the same results would hold on FC100 and TieredImageNet. The latter benchmarks are specifically designed to increase task diversity, and I wonder if the proposed metric would be able to quantify this diversity. Second, I would like to see a comparison of methods to measure task diversity. Three come to mind:
        - First, the authors could use the ones they mention (eg, CCA, CKA) to measure similarity between the obtained representations.
        - Second, Dhillon et al., 2019 propose a measure of task difficulty, which can be used to measure diversity (along the axis of difficulty).
        - Third, Ethayarajh et al., 2022 (Understanding Dataset Difficulty with V-Usable Information) also propose a measure of difficulty between datasets, but this measure actually builds on the information contained in a dataset (akin to the diversity in this paper).
    - Missing discussion of prior art: several important prior works are missing from the discussion in this paper. In its current form, I am not sure that the paper offers novel insights.
        - Dhillon et al., 2019 shows the same results as Tian et al., 2020, and so does Wang et al., 2019 (SimpleShot: Revisiting Nearest-Neighbor Classification for Few-Shot Learning).
        - Arnold and Sha, 2021 (Embedding Adaptation is Still Needed for Few-Shot Learning) also show that transfer learning methods shine when train and test tasks are similar, but MAML has the upper hand whenever they are dissimilar. Collins et al., 2020 (How Does the Task Landscape Affect MAML Performance?) compare the settings under which MAML is competitive and can outperform a pretrained feature extractor.
        - Several recent works (incl. at ICLR this year) have carefully studied MAML and drastically improved its performance on few-shot classification. See for example: Ye and Chao, 2022 (How to Train Your MAML to Excel in Few-Shot Classification).
    - Presentation: the paper needs significant work in terms of presentation to be up to the ICLR standards. Some examples:
        - Equation 2 and 3 have formatting issues.
        - Citation: Achille UCLA et al., 2019 should not have UCLA.
        - Figure 1 needs improvements to be legible. It is also misleading — in few-shot classification, we typically use 600 or 2000 tasks to compute confidence intervals. Doing so would significantly shrink the error bars.

**Summary Of The Paper:**

This paper studies when and how meta-learning algorithms should be used to solve few-shot classification tasks. The authors introduce a “diversity coefficient”, which they use to measure how varied a set of tasks is. They show that (some) standard few-shot classification have low diversity — in other words, all tasks are easy so both MAML and transfer learning perform equally well. From those experiments, they conclude that meta-learning and transfer learning methods yield equivalent performance on few-shot classification tasks.

**Summary Of The Review:**

- Main strength:
    - Use of Task2Vec to measure task similarity in few-shot classification.
- Main weakness:
    - Lack of novelty — most insights in the paper are either well-known or obsolete by now.

---

### Official Review · Reviewer_Bxys · 2022-10-26

**Confidence:** 4
**Correctness:** 3
**Technical Novelty And Significance:** 3
**Empirical Novelty And Significance:** 3
**Recommendation:** 6

**Clarity, Quality, Novelty And Reproducibility:**

The paper is mostly clear although it could be improved (see above).
Results are novel and seem reproducible.

**Strength And Weaknesses:**

Strengths

- The debate around whether transfer learning or meta-learning should be used for few-shot classification is interesting and not resolved.
- Experiments are convincing: they run with multiple architectures, model sizes, datasets, metrics and have several controls.


Weaknesses

- Showing differences between USL and MAML for high task diversity is crucial for testing the central hypothesis of this paper.
Yet, this is done only in synthetic Gaussian data in section 4.5.
It would be good to have a high diversity benchmark with real data, instead of Gaussians.
This could be done, for example, by separating the 100 classes of CIFAR-FS into hierarchies, with separate tasks for animals, vehicles, etc.
One way of doing this is described in https://ojs.aaai.org/index.php/AAAI/article/view/20590

- I find it very surprising that USL is better than MAML for high task diversity.
MAML is designed to deal precisely with this situation, while USL is not.
The authors should discuss this result in much more detail.


Minor points

- I don't understand why Eq.2 needs to be in this paper, it doesn't seem to be used anywhere.

- "distances based on Task2Vec are good approximations to the ground truth distance of task distribution."
I don't understand this sentence, FIM is a metric in the space of parameters (or parametric distribution), not in the space of tasks.
What is the "ground truth distance of task distribution" anyway?
Overall, section 3.3 includes several unclear (and maybe wrong) statements, this section could be entirely removed.

- Results in Table 2 are not entirely clear to me.
How can USL perform worse than chance (<20%) in any circumstance?
Why results depend so much on the adaptation method used?

- "the two methods are equivalent in a statistically significant way".
This statement is repeated several times in the paper but is incorrect.
The correct statement is: "differences in the two methods are not statistically significant"



**Summary Of The Paper:**

This paper studies the question of whether transfer learning (USL) is superior to meta-learning (MAML) for few-shot classification (multi-task), as suggested by recent studies.
It shows that the two methods have identical performance when task diversity is low, while their performance is different when task diversity is high.
It proposes a new measure of task diversity and argues that previous studies had not compared the two methods fairly.
It also shows that predictions of USL and MAML are similar although they learn different representations.



**Summary Of The Review:**

Overall the strengths slightly overweight the weaknesses, but the paper could be improved a lot.

---

### Decision · Program_Chairs · 2023-01-20

**Decision:**

Reject

**Justification For Why Not Higher Score:**

The limits of the experimental study and discussion with state of the art are too important.

**Justification For Why Not Lower Score:**

N/A

**Metareview: Summary, Strengths And Weaknesses:**

The paper addresses the problem of few-shot learning and studies the performance of meta-learning and transfer learning for this problem.

Among the strengths the paper, we can mention that the problem studied is interesting, the diversity criterion considered is interesting and the experiments have interesting aspects.

However, the reviewers unanimously recognized that the novelty of the paper is not enough, the experimental study lacks of a larger evaluation and some important related are not sufficiently discussed.

This paper still needs some work. I propose then rejection.